# Overview of Mpox Outbreak in Greece in 2022–2023: Is It Over?

**DOI:** 10.3390/v15061384

**Published:** 2023-06-16

**Authors:** Kassiani Mellou, Kyriaki Tryfinopoulou, Styliani Pappa, Kassiani Gkolfinopoulou, Sofia Papanikou, Georgia Papadopoulou, Evangelia Vassou, Evangelia-Georgia Kostaki, Kalliopi Papadima, Elissavet Mouratidou, Maria Tsintziloni, Nikolaos Siafakas, Zoi Florou, Antigoni Katsoulidou, Spyros Sapounas, George Sourvinos, Spyridon Pournaras, Efthymia Petinaki, Maria Goula, Vassilios Paparizos, Anna Papa, Theoklis Zaoutis, Dimitrios Paraskevis

**Affiliations:** 1National Public Health Organization, 15123 Athens, Greece; k.tryfinopoulou@eody.gov.gr (K.T.); k.gkolfinopoulou@eody.gov.gr (K.G.); sopavage@yahoo.gr (S.P.); georgiapa2812@gmail.com (G.P.); evassou.7@gmail.com (E.V.); k.papadima@eody.gov.gr (K.P.); e.mouratidou@eody.gov.gr (E.M.); icu.rec2@eody.gov.gr (M.T.); a.katsoulidou@eody.gov.gr (A.K.); s.sapounas@eody.gov.gr (S.S.); t.zaoutis@eody.gov.gr (T.Z.); dparask@med.uoa.gr (D.P.); 2Central Public Health Laboratory, 16672 Athens, Greece; 3Department of Microbiology, Medical School, Faculty of Health Sciences, Aristotle University of Thessaloniki, 54124 Thessaloniki, Greece; s_pappa@hotmail.com (S.P.); annap.med@gmail.com (A.P.); 4Department of Hygiene, Epidemiology and Medical Statistics, Medical School, Kapodistrian University of Athens, 11527 Athens, Greece; ekostakh@med.uoa.gr; 5Clinical Microbiology Laboratory, Attikon General University Hospital of Athens, 12462 Athens, Greece; nsiaf@med.uoa.gr (N.S.); s.pournaras@med.uoa.gr (S.P.); 6Department of Medical Biopathology, Faculty of Medicine, University of Thessaly, 41500 Larissa, Greece; zoi_fl@yahoo.gr (Z.F.); petinaki@uth.gr (E.P.); 7Laboratory of Clinical Virology, School of Medicine, University of Crete, 71003 Heraklion, Greece; sourvino@med.uoc.gr; 8State Dermatology Department, Hospital of Skin and Venereal Diseases, 54643 Thessaloniki, Greece; drmgoula@gmail.com; 91st Department of Dermatology and Venereology, National and Kapodistrian University of Athens Medical School, “Andreas Syggros” Hospital for Skin and Venereal Diseases, 16121 Athens, Greece; vpaparizos@yahoo.gr

**Keywords:** orthopoxvirus, mpox, virus, outbreak, response, vaccination

## Abstract

In May 2022, for the first time, multiple cases of mpox were reported in several non-endemic countries. The first ever case of the disease in Greece was confirmed on 8 June 2022, and a total of 88 cases were reported in the country until the end of April 2023. A multidisciplinary response team was established by the Greek National Public Health Organization (EODY) to monitor and manage the situation. EODY’s emergency response focused on enhanced surveillance, laboratory testing, contact tracing, medical countermeasures, and the education of health care providers and the public. Even though management of cases was considered successful and the risk from the disease was downgraded, sporadic cases continue to occur. Here, we provide epidemiological and laboratory features of the reported cases to depict the course of the disease notification rate. Our results suggest that measures for raising awareness as well as vaccination of high-risk groups of the population should be continued.

## 1. Introduction

Human mpox is a viral zoonosis caused by the mpox virus that was first recognized in 1958 when monkeys that had been shipped from Singapore to a Denmark research facility fell ill [1]. Mpox virus is a double-stranded DNA virus and a member of the orthopoxvirus genus within the Poxviridae family.

The first human case was reported in the Democratic Republic of Congo in 1970, when the virus was isolated from a child suspected to have smallpox [2]. The coincident immunity to the mpox virus previously achieved with vaccinia vaccination decreased over the years as the eradication of smallpox and subsequent lack of vaccination efforts paved the way for the virus to gain clinical relevance [3]. Up to May 2022, cases had been recorded in the tropical regions of West and Central Africa as a result of transmission through contact with wild animals or from human to human through close physical contact [4].

The first mpox outbreak outside the African continent was detected in the United States in May 2003 and was linked to contact with infected pet prairie dogs. This outbreak resulted in over 70 mpox cases [5,6]. These pets had been housed with Gambian pouched rats and dormice that had been imported from Ghana. Since then, imported cases have been reported in Israel (2018), United Kingdom (2018, 2019, 2021, 2022), Singapore (2019), and the USA in July and November 2021 [7,8,9,10,11,12,13].

Person-to-person transmission mainly occurs through contact with the skin lesions/bodily fluids of the infected person. The virus can enter through the mucous membranes and small wounds/cracks in the skin.

The virus’s initial symptoms include fever, headache, myalgia, fatigue, and lymphadenopathy, a key differentiating feature of mpox from smallpox. After 1 to 2 days, mucosal lesions develop in the mouth, closely followed by skin lesions of the face and extremities (including palms and soles), and are centrifugally concentrated. The rash may or may not spread to the rest of the body, and the total number of lesions may vary from a small amount to thousands [14].

In May 2022, multiple non-travel-related cases of mpox were reported for the first time in European countries, mainly among men having sex with men (MSM) [15]. On 23 July 2022, the Director General of the WHO declared the multi-country outbreak of mpox to be a Public Health Emergency of International concern (PHEIC) [16]. In total, until 4 April 2023, 25,874 laboratory confirmed cases of the disease were recorded in 45 European Union/European Economic Area (EU/EEA) countries, including six deaths [17].

Within one week of the report of the first suspected case of mpox in Greece on June 8, the National Public Health Organization (EODY) launched an emergency response plan for mpox. We present here the epidemiological and laboratory features of identified mpox cases and describe the response of the Greek public health authorities, depicting main lessons learned and providing evidence for future public health strategies against the disease.

## 2. Materials and Methods

### 2.1. Surveillance and Epidemiological Investigation of Cases

An enhanced mpox surveillance system was implemented. A notification form, an investigation form, and a form for recording and following up close contacts of possible/confirmed cases were created. The ECDC definitions for probable and confirmed cases were used.

All reported mpox cases were investigated. The investigation was performed via communication with the attending physicians and/or via interviews with the patients. These were conducted by specially trained personnel using a structured epidemiological questionnaire. The data collected included sociodemographic, clinical, and behavioral data.

Data were used to describe cases in terms of a person’s characteristics (age, sex), clinical manifestations and disease severity, probable place of exposure, time, and putative risk factors (i.e., recent travel, contact with other confirmed or probable cases, sexual orientation, recreational activities, contact with animals, and HIV status). The number of cases by ISO (International Organization for Standardization) week depicts the temporal distribution of the recorded cases in the country. The ISO weeks system uses the week containing the first Thursday of the year as the first week of the year, and weeks start with Monday and end on Sunday.

The exposure investigation covered the period of 21 days prior to symptom onset.

A contact tracing team was formed for the identification and follow-up of close contacts, aiming to interrupt transmission and support people at a higher risk of developing severe disease. Cases were prompted to identify contacts across a number of contexts, such as household, workplace, sexual contacts, healthcare, sports, social gatherings, and any other recalled interactions.

Contacts were notified of their exposure (preferably by the case) and asked to self-monitor daily for the onset of signs or symptoms for a period of 21 days following the last case of contact with the probable or confirmed case or their contaminated materials during the infectious period.

Actions triggered by the notification of a suspected mpox case are summarized in Figure 1.

In parallel, close monitoring of data from other countries inside and outside Europe for continuous risk assessment was performed.

The EODY suggested the inclusion of the disease in the mandatory notification system. However, the update of the list of mandatory notifiable diseases in the respective law is pending.

### 2.2. Laboratory Investigation

As a range of conditions can cause skin lesions and clinical presentation of mpox cases was not always typical, laboratory-based diagnosis was of paramount importance for the management of the outbreak.

The Central Public Health Laboratory (CPHL) in Athens, the National Reference Lab for emerging viruses in Thessaloniki, Northern Greece, and the Clinical Virology Laboratory in Heraklion, Crete were appointed by the EODY to serve as the main laboratories for the confirmation of suspected cases. During the course of the outbreak, samples were also analyzed by the clinical microbiology laboratories of Attikon General University Hospital in Athens and the University Hospital of Larissa. These institutions developed the capacity to test samples from visiting patients that met the criteria for testing.

The recommended skin lesion material, including swabs of lesion surface or exudate and lesion crusts, was collected from each case. DNA was extracted using a commercial manual extraction kits (QIAamp DNA Mini Kit from QIAGEN, Hilden, Germany or genesig Easy, Primerdesign, Southampton, UK).

Initially, during June 2022, a generic orthopoxvirus real-time PCR assay (RealStar orthopoxvirus PCR kit, altona Diagnostic, Hamburg, Germany), which targets genes common to other orthopoxviruses (e.g., smallpox, vaccinia), was used as a screening test or an in-house PCR protocol using the OPS3 and OPAs4 primers described by Panning et al. [18], along with the CDC non-variola orthopox generic real-time PCR and subsequent Sanger sequencing. However, since early July, all the laboratories have been used a monkeypox-specific, real-time PCR test for the qualitative identification of DNA from monkeypox virus in clinical samples (Viasure Monkeypox virus Real Time PCR Detection kit, Zaragosa, Spain).

Whole-genome sequencing analysis was performed in two samples. DNA was extracted from skin swab lesions samples by using the High Pure Viral Nucleic Acid Kit (Roche Life Sciences, Basel, Switzerland). The extracted DNA was quantified using a Qubit dsDNA High-Sensitivity Assay kit (Invitrogen, Thermo Fisher Scientific, Waltham, MA, USA) following the protocol of use as defined by the manufacturer’s guidelines. One μg of DNA was used to prepare the library with the ligation Sequencing Kit- SQK-LSK109 and the Native Barcoding Expansion 1–12 (PCR-free) EXP-NBD104 (Oxford Nanopore Technologies, Oxford, UK) following the manufacturers’ protocol. The final library (~430 ng) was loaded onto a FLO-MIN106 R9.4.1 flow cell on a MinIon Mk1C device using fast accuracy basecalling. The genome was assembled with Geneious v. 7.1.3 (https://www.geneious.com, accessed on 20 March 2023) using the MPXV sequence with GenBank accession number NC_063383 as reference.

The two sequences sampled in Greece were analyzed phylogenetically along with 36 reference sequences representative of all lineages (B.1, B.1.1, B.1.10, B.1.11, B.1.12, B.1.13, B.1.14, B.1.15, B.1.16, B.1.17, B.1.2, B.1.3, B.1.4, B.1.5, B.1.6, B.1.7, B.1.8, and B.1.9) sampled during the period from1 May 2022 and 29 September 2022. Specifically, two high-quality sequences per lineage were selected and downloaded from the GISAID database [19] so as to make the sampling window per lineage at least 52 days (1.7 months). Multiple-sequence alignment was performed using the MAFFT program (The CIPRES Science Gateway, version 3.3; https://www.phylo.org/, accessed on 20 March 2023). Phylogenetic analysis was performed with the HKY + F + I as an evolutionary model, and an ultrafast bootstrap with 1000 replicates using the IQ-TREE, version 2.1.2 [20]. The phylogenetic tree was visualized using the FigTree (version 1.4) program.

### 2.3. Ethical Considerations

The EODY is authorized by the Greek law to process COVID-19 epidemiological data for public health purposes. No personal data were used. The study was conducted in accordance with the national and European Union regulations and approved by the Institutional Review Board of the EODY (April 2023).

## 3. Results

### 3.1. Epidemiology

The first suspected monkeypox case in Greece was initially reported on 8 June 2022. By 20 April 2023, 88 cases were reported. Among this group, six were travelers visiting Greece and the rest were Greek residents from 6 out of 13 NUTS-II regions (49 from the Attica region). All 88 cases were men: 82 of them (93.2%) self-identified as MSM. Cases median age was 37 years (16–62 years); all but one patient was an adult.

The time distribution of laboratory-confirmed cases by week of symptom onset (ISO week) is shown in Figure 2.

The epidemic curve is compatible with person-to-person transmission up to week 38. However, even though the outbreak was typically over in the country at around week 41, sporadic mpox cases were reported in weeks 44 and 52 of 2022 and weeks 5 and 16 of 2023. These patients had not travelled outside the country during the incubation period. Their ages ranged between 23 and 62, and two of them self-identified as heterosexual.

Table 1 summarizes cases’ characteristics. Seventy-three cases were Greek (82.9%). Twenty-four cases reported travel to a country that had reported mpox cases during the incubation period (27.3%). Sexual contact was reported as the most likely way of transmission for 78 cases, 76 (97.4%) of them being self-identified as MSM. Of 88 cases, 13 reported as having attended a large event with sexual contact or having sex in the context of parties (night club/private party/sauna or similar setting) before symptom onset. Among 37 cases with known HIV status, 26 (70.2%) were HIV-positive. Eight cases were health care workers.

The most frequently reported symptom was rash (88, 100%), followed by lymphadenopathy (69, 73.9%) and fever (65, 73.9%). A synchronous evolution of the skin rash was recorded in 52 cases (59.1%). The most frequent localization of the rash was the genital area (61.4%).

In total, fifteen cases (17.1%) were hospitalized, whereas no intensive care unit admissions were reported, nor were deaths. Out of 84 cases with known smallpox vaccination status, only one case (1.2%) was vaccinated.

### 3.2. Laboratory Findings

Of 238 suspected cases, 88 (52.7%) tested positive for MPXV DNA when assessed by PCR, with confirmation by Sanger sequencing in the first cases. From early July, the availability of an mpox-specific, real-time PCR test facilitated the confirmation of the suspected cases in the same day of sample arrival to the laboratories ensuring early diagnosis.

As for sequence analysis obtained by analyzing the complete genome, a total of 372,658 raw reads were taken from the sample MPXV-GR-386-2022 (collected on 22 July 2022), with sizes ranging from 60 bp to about 42 kb, and 99.8% (39,026 reads) were assigned for the assembly, presenting 72 nucleotide substitutions. This resulted in 31 amino acid changes. For sample MPXV-GR-613-2022 (collected on 22 August 2022), a total of 486,023 raw reads were taken, with sizes ranging from 38 bp to about 24 kb, and 99.9% (27,997 reads) were assigned for use in the assembly process, presenting 71 nucleotide substitutions, which resulted in 32 amino acid changes.

Both sequences belong to clade IIb of B.1 lineage (Nextclade v2.13.0), which is consistent with other sequences from the 2022 hMPXV outbreak.

Phylogenetic analysis performed using sequences from all previously described lineages suggested that one sequence was classified as B.1.1. and that the other one was more closely related to B.1.11, although this was with a long branch (Figure 3).

The data for this study have been deposited in the European Nucleotide Archive (ENA) at EMBL-EBI under accession number PRJEB61234.

## 4. Response

A multi-disciplinary response team was formed to set the priorities towards this emerging public health threat. Public health specialists, epidemiologists, clinicians, veterinarians and representatives of non-governmental organizations collaborated for the effective response. A main consideration of the team was to protect the public, without discriminating against MSM or other groups of the population.

The basic pillars of action were to provide guidance for clinicians and the public and to implement control and prevention measures to minimize public health consequences. Information was sent to hospitals, health care centers and medical associations informing of the occurrence of cases, basic clinical manifestations and instructions on protection and hygiene measures. The importance of strict adherence to standard contact precautions, hand hygiene and barrier nursing through use of Personal Protective Equipment (PPE) including: gloves, face mask, gown and goggles, was stressed.

Patients with suspected mpox had to be isolated during the presumed and known infectious periods, that is during the prodromal and rash stages of the illness, respectively. Isolation continued until all of the lesions had resolved.

It was stressed that implementation of measures should not be delayed in anticipation of laboratory confirmation and all probable cases were treated as confirmed until laboratory investigation showed otherwise.

Guidance was provided on clinical management of possible and confirmed mpox cases. The European civil protection mechanism (rescEU) was successfully mobilized for treating a patient presented with non-healing corneal ulcer, who has fully recovered after receiving oral tecovirimat [21].

Finally, after the first few weeks of the outbreak the necessity to have an approval system for testing of samples was identified. Clinical sings and symptoms of mpox are not specific and it in several cases it was difficult to distinguish mpox on the basis of clinical presentation alone, especially for cases with an atypical appearance. As various conditions may cause skin rashes, it was considered important to rule out other potential causes of different skin lesions or rash before testing (i.e., herpes simplex virus, varicella zoster virus, molluscum contagiosum virus, enterovirus, measles, scabies, syphilis, bacterial skin infections, rickettsia pox, drug allergies). The main reason for setting this approval system was to assure the laboratory capacity of the country and avoid unnecessary waste of resources. Thus, a trained team at the central level discussed with the physician about the clinical manifestations, demographics, and epidemiological history of each reported suspected case before approving testing at the reference laboratories. Testing for varicella infection (especially for children) and herpes was requested before testing for mpox. Decision on testing was based on both clinical and epidemiological factors, linked to an assessment of the likelihood of the case to actually being infected.

Standard operating procedures for biosafety issues as well as for samples’ collection, storage and transportation to the reference laboratories have been issued by the CPHL and uploaded to the EODY site.

In parallel, actions for the public took place. The EODY issued a press release and uploaded Frequently Asked Questions at its official website, with the intention to minimize the effect of fake news spreading online. Efforts were made to reach people experiencing the highest risk of infection (via on site visits at gay bars, festivals, etc.). Organizations, including activist groups and community testing organizations (i.e., checkpoints) working on health for MSM and lesbian, gay, bisexual, and transgender (LGBT) communities, were contacted and requested to participate with their members and networks in raising awareness of the public.

A dedicated mission was organized in a highly popular island touristic destination, aiming at sensitizing high risk groups, especially in view of the summer 2022 touristic season. Personnel of the health care centers on the islands were informed on the signs and symptoms they needed to be in alert for and were offered support for clinical management of suspected cases and diagnosis. In addition, a hotline was established by the EODY, where both the public and health care professionals could address questions about the disease.

Finally, the necessary actions were taken for the national provision of JYNNEOS vaccines, and the country created a vaccine stock. Instructions for the disposal of this were finalized by the National Immunization Committee.

## 5. Discussion

In May 2022, non-travel related cases of mpox were reported to the WHO from several non-endemic countries. In the following, weeks the number of cases continued to increase worldwide, leading to a global outbreak. Early cases were mostly travel-related. However, by the end of May, locally acquired infections and community transmission became predominant in all affected countries [22]. Epidemiological data suggested the association of the early reported cases of the disease with an international LGBT+ Pride event on the island of Gran Canaria. These cases were then linked to transmission chains in several European countries [23,24,25].

In Greece, the burden of disease was lower compared to other European countries. However the course of the outbreak was comparable to the course in other European countries. Cases continue to be reported sporadically, with the last case reported on the 20th of April 2023. Further study is needed to investigate whether this is a result of ongoing transmission in the community that is not being captured by the surveillance system. Studies on the percentage of asymptomatic cases of the disease are needed in order to obtain a clearer picture of the epidemiology of the disease.

Regarding the epidemiological characteristics of the reported cases, data were compatible with data from other countries [18]. Cases were all young male adults. A relatively high percentage of HIV-positive cases was recorder among cases with known HIV status, which underlines the need to test cases diagnosed with mpox for HIV and other sexually transmitted infections as well, taking into consideration that people with untreated HIV are more likely to have complications of mpox [26]. Even though in the literature the highest occupational risk associated with the mpox virus appears to apply to healthcare professionals (physicians, nurses, nursing assistants, emergency, medical technicians, therapists, pharmacists, students, laboratory workers) [27,28], no cases were identified among health care professionals that treated cases in Greece, despite the fact that they were not vaccinated against the virus.

Most cases identified themselves as MSM. Clinical presentation involved predominantly genital, perineal or perianal lesions, with inguinal lymphadenopathy sexual transmission being the main mode of transmission, as described in other studies [22,29].

Historically, in terms of its genomic and evolutionary characteristics, two distinct clades of MPXV had been identified in different geographical regions of Africa. The “Central African” clade, with a case-fatality rate of 1–12%, found in central Africa and the Congo Basin, and the less virulent “West African” clade, with case-fatality rate of less than 0·1%, that is found in west Africa [30,31]. However, Happi et al. [32] proposed a novel classification of MPXV that is non-discriminatory and non-stigmatizing for specific nations and geographic regions and this has been accepted by the WHO. According to the new classification, clade I corresponds to the prior “Central African clade”, while clade II, with its two subgroupings IIa and IIb, corresponds to the prior “West African clade”. Subclade IIb includes genomes associated with the most recent outbreaks in humans (hMPXV), including the current MPXV outbreak. Moreover, the different outbreaks occurring in recent years have reshaped the genomic landscape of subclade IIb, leading to a further divergence of lineages (A, A.1, A.1.1, A.1.1, B.1) [33]. The new lineage B.1 of clade IIb has been identified in the 2022 global outbreak [31,33] and up to now several different B.1.1 lineages have been recognized. The two MPXV strains from Greece fall into B.1.1 and B.1.11 lineages. Although most B.1 lineages appear to be widely distributed, some are predominantly present only in certain countries. For instance, the majority of B.1.6 genomes originate from Peru, B.1.4 from Canada and B.1.11 mostly from the USA [33].

We consider that the level of preparedness and the response applied to the international signals were successful, as diagnostic laboratory capacity was adequate from the early stages of the outbreak, clinical doctors were sensitized, and high-risk groups of the population were informed. Special efforts during summer, in an extremely high touristic season for Greece, are deemed to be successful given the fact that the number of cases reported from popular islands that hosted events for the MSM community was not high. In parallel, we did not receive any reports from EWRS/IHR about tourists travelling to Greece having been infected by the disease. On the other hand, the low number of reported cases may also be attributed to an underreporting of probable cases to a degree that is not easy to estimate.

To further strengthen the disease surveillance, the necessary steps to integrate mpox into the mandatory notification system have been taken, with the practice rooted in clinicians—especially targeting those who do not work directly in STI clinics—of the need to be aware of mpox symptoms and the possibility that cases may reappear.

A main challenge in the response was contact tracing, as most of the cases could not share contact details of their sexual partners, either because they did not have this information, or they did not want to share with public health authorities. Partner notification was challenging and required caution so that personal data were kept confidential.

Another challenge in the management of the outbreak was that testing criteria were not always met and, in the earlier stage of the outbreak, samples were tested even when the pre-test probability to actually be positive was extremely low (e.g., samples from children without travel history or possible exposure to a case that finally tested positive for varicella infection). Systematic effort should be made to reinforce testing where it is most needed, targeting clinical settings that serve the population at highest risk. The possibility of integrating mpox testing into existing HIV and STI programmes should be explored.

The identification of sporadic domestic cases in Greece and other countries [17] after the end of the outbreak depicts the need to strengthen surveillance and retain a well-coordinated plan to prevent a resurgence of cases, focusing mainly on raising awareness among high-risk population groups, preparing health care units, and assuring testing availability especially as the new summer high touristic season for Greece approaches.

## Figures and Tables

**Figure 1 viruses-15-01384-f001:**
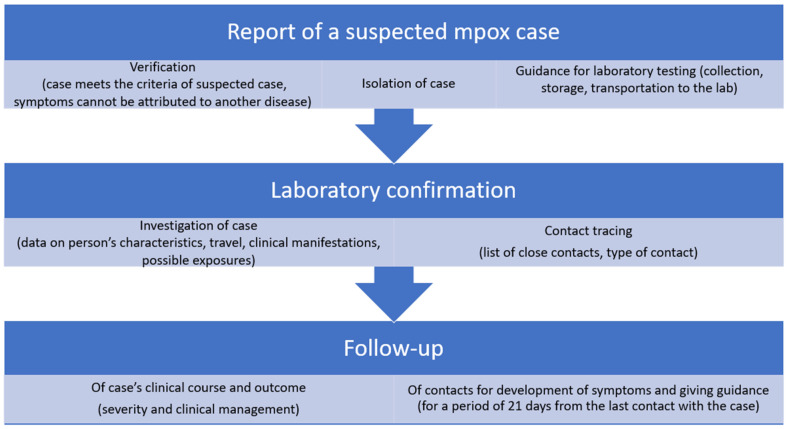
Actions triggered by the notification of a suspected mpox case.

**Figure 2 viruses-15-01384-f002:**
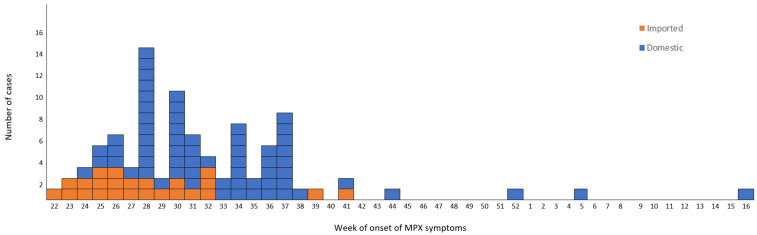
Number of confirmed cases of Mpox by week of symptom onset in Greece, from 2 June 2022 to 18 April 2023.

**Figure 3 viruses-15-01384-f003:**
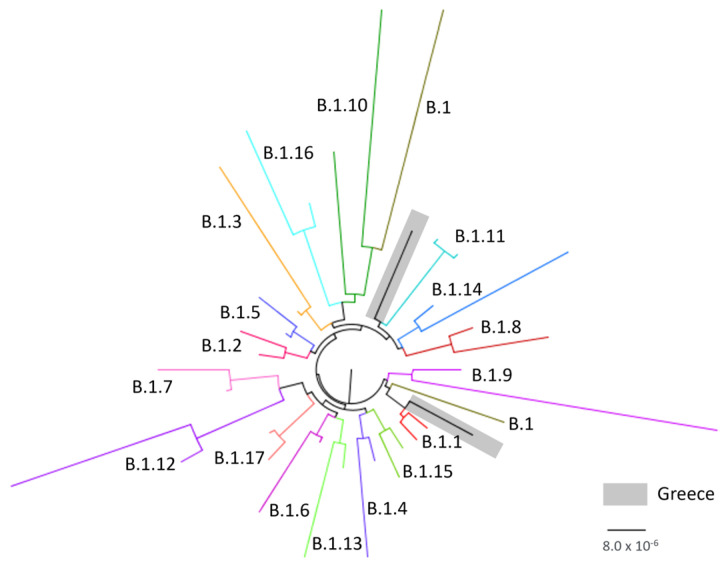
Phylogenetic tree of Greek MPXV genomes using reference sequences representative of all lineages from GISAID database.

**Table 1 viruses-15-01384-t001:** Demographic, epidemiological and clinical characteristics of mpox cases, Greece, 2 June 2022 to 18 April 2023.

Characteristic	n	%
**Age groups, yrs**		
0–17	1	1.1
18–30	24	27.3
31–40	33	37.5
41–50	23	26.1
51–60	5	5.7
>60	2	2.3
**Gender**		
Male	88	100.0
**Place of residence (NUTS II Region)**		
Regional Area of Attica	49	55.7
Central Macedonia	22	25.0
Crete	4	4.5
Thessaly	4	4.5
South Aegean Sea	2	2.3
Ionian Islands	1	1.1
Travelers/Tourists	6	6.8
**Nationality**		
Greek	73	83.0
Other	15	17.0
African	1	1.1
American	2	2.3
Asia	3	3.4
Europe	9	10.2
**HIV status**		
Negative	11	12.5
Positive	26	29.5
Unknown	51	58.0
**Travel history during the incubation period**		
Yes	24	27.3
Νο	64	72.7
**Possible route of transmission**		
Sexual contact	78	88.6
Unknown	10	11.4
**Prior smallpox vaccination**		
Vaccinated	1	1.1
Not vaccinated	83	94.3
Unknown	4	4.6
**Clinical symptoms**		
Rash	88	100.0
Lymphadenopathy	69	78.4
Fever	65	73.9
Fatigue	52	59.1
Headache	34	38.6
Myalgia	32	36.4
Back pain	23	26.1
Itching	5	5.7
Pain	3	3.4
**Evolution of skin lesions**		
Synchronous	52	59.1
Asynchronous	33	37.5
Unknown	3	3.4
**Localisation of the skin lesions**		
Genital	54	61.4
Arms	24	27.3
Face	22	25.0
Anal-perianal	20	22.7
Legs	14	15.9
Mouth	13	14.8
Trunk	9	10.2
Thorax	9	10.2
Palms	5	5.7
Soles	4	4.5
Head	2	2.3

## Data Availability

Data available upon request.

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
