# Peer review of "Overview of Mpox Outbreak in Greece in 2022–2023: Is It Over?"

_viruses, 2023, doi:10.3390/v15061384_

Round 1

Reviewer 1 Report

The authors presented epidemiological ,clinical and laboratory data regating the MPX outbreak in Greece 2022-23. The manuscript is quite clear. Moderate English editing is required.

I suggest to add a figure in order to illustrate with an algotrithm the epidemiological process that can be provided to increase the contact tracing, the MPX diagnosis and linkage to care.

Moderate editing

Author Response

Dear editor

thank you for your review.

We followed your suggestion to add a figure to illustrate the  process that was actually adopted for the management of suspected mpox cases.

It is now included in the revised manuscript.

Reviewer 2 Report

The manuscript by Mellow et al. is dedicated to mpox outbreak in Greece in 2022-2023. The main focus of the manuscript is on the epidemiology of mpox, and the control of mpox outbreaks in Greece in 2022-2023. A multidisciplinary response team was established by the Greek National Public Health Organization to monitor and manage mpox outbreak in Greece.

It was also found that of the two sequenced samples from Greece, one sequence was classified as B.1.1. and the other one was more closely related to B.1.11. Overall, this is quite a well-written and well-designed paper. I believe that this manuscript can be published in the Viruses if the epidemiology and disease control correspond to the subject of the journal.

Specific comments:

Page 2: There is no parenthesis at the end of the sentence “(Viasure Monkeypox virus Real Time PCR Detection kit, Spain.:

Page 3: What does “ISO week” mean? Are there different weeks?

Author Response

Dear reviewer thank you for your review.

Your comments were addressed as follows:

Page 2: There is no parenthesis at the end of the sentence “(Viasure Monkeypox virus Real Time PCR Detection kit, Spain.:

Thanks. The parenthesis was added.

Page 3: What does “ISO week” mean? Are there different weeks?

Thank you for this comment. Your are right. This is a technical term and readers may not be familiar with this.

ISO weeks are commonly used in epidemiology to deal with the fact that every year starts from different day of the week thus the number of weeks differ from year to year.

The ISO week date system is effectively a leap week calendar system that is part of the ISO 8601 date and time standard issued by the International Organization for Standardization (ISO). An ISO week-numbering year has 52 or 53 full weeks. That is 364 or 371 days instead of the usual 365 or 366 days. Weeks start with Monday and end on Sunday. 

ISO Week system uses the week containing the first Thursday of the year as the first week of the year. 

For making this point clear we added a sentence in the methods section as follows:

"The number of cases by ISO week depicts the temporal distribution of the recorded cases in the country. ISO weeks system uses the week containing the first Thursday of the year as the first week of the year and weeks start with Monday and end on Sunday".